# The RETurn to work After stroKE (RETAKE) trial: Findings from a mixed-methods process evaluation of the Early Stroke Specialist Vocational Rehabilitation (ESSVR) intervention

David James Clarke[1], Katie Powers[2], Diane Trusson[2], Kristelle Craven[2], Julie Phillips[2], Jain Holmes[2], Christopher McKevitt[3], Audrey Bowen[4], Caroline Leigh Watkins[5], Amanda J. Farrin[6], Alexandra Wright-Hughes[6], Tracey Sach[7], Rachel Chambers[3], Kate Radford[2]*

1 Academic Unit for Ageing and Stroke Research, Leeds Institute of Health Sciences, University of Leeds, Leeds, United Kingdom, 2 Centre for Rehabilitation & Ageing Research Medicine/ Injury, Inflammation and Recovery Sciences, School of Medicine, Medical School Queen's Medical Centre, University of Nottingham, Nottingham, United Kingdom, 3 Department of Population Health Sciences, Faculty of Life Sciences & Medicine, King's College London, London, United Kingdom, 4 Manchester Centre for Health Psychology & the Geoffrey Jefferson Brain Research Centre MAHSC, University of Manchester, Manchester, United Kingdom, 5 Applied Health Research Hub, Stroke Research Team, School of Nursing and Midwifery, University of Central Lancashire, Preston, United Kingdom, 6 Clinical Trials Research Unit (CTRU), Leeds Institute of Clinical Trials Research, University of Leeds, Leeds, United Kingdom, 7 Health Economics Group, Norwich Medical School, Norwich Research Park, University of East Anglia, Norwich, United Kingdom

* kate.radford@Nottingham.ac.uk

## Abstract

### Introduction

A key goal for working age stroke survivors is to return to work, yet only around 50% achieve this at 12 months. Currently, there is limited evidence of effectiveness of early stroke-specialist vocational rehabilitation (ESSVR) interventions from randomised controlled trials. This study examined fidelity to ESSVR and explored social and structural factors which may have influenced implementation in the RETurn to work After stroKE (RETAKE) randomised controlled trial.

### Methods

Mixed-methods process evaluation assessing intervention fidelity and incorporating longitudinal case-studies exploring stroke survivors' experiences of support to return to work. Normalisation Process Theory, and the Conceptual Model for Implementation Fidelity, informed data collection and analysis.

### Results

Sixteen sites across England and Wales participated in RETAKE. Forty-eight occupational therapists (OTs), supported by 6 mentors experienced in vocational rehabilitation (VR), delivered the intervention (duration 12 months) between February 2018 and April 2022. Twenty-six participants (15 ESSVR, 11 usual care (UC)) were included in longitudinal case-

**Data Availability Statement:** There are some restrictions on public access to data included in this manuscript. These restrictions relate to the

consent provided by the participants in the study for access to data they provided for the study. The Health Research Authority (HRA) for the United Kingdom (UK) and the Research Ethics Committee (REC) East Midlands - Nottingham 2 approved the study. For the RETAKE study, including the embedded Process Evaluation reported in this manuscript, the University of Nottingham and the University of Leeds are joint data controllers (legally responsible for the data security) and the Chief Investigator of this study Professor Kate Radford is the Data Custodian (manages access to the data). Reasonable requests for access to data reported in the Process Evaluation may be sent to RETAKE@leeds.ac.uk for access to the dataset. Access will be governed by an information governance committee formed between the University of Nottingham and the University of Leeds.

**Funding:** This study was funded by the NIHR HTA programme (ref: 15/130/11). The views expressed herein are those of the authors, not necessarily the NIHR, the Department of Health and Social Care or the NHS. The Funders had no role in study design, data collection, and analysis, decision to publish or preparation of the manuscript.

**Competing interests:** The authors have declared that no competing interests exist.

studies. An additional 18 participants (8 ESSVR and 10 UC) were interviewed once. Nineteen OTs, 6 mentors and 19 service managers were interviewed. Fidelity was measured for 39 ESSVR participants; mean fidelity score was 78.8% (SD:19.2%, range 31–100%). Comparison of the experiences of ESSVR and UC participants indicated duration and type of support to return to work were perceived to be better for ESSVR participants. They received early, co-ordinated support including employer liaison and workplace adjustments where appropriate. In contrast, UC participants reported limited or no VR or return to work support from health professionals. Typically, UC support lasted 2–8 weeks, with poor communication and co-ordination between rehabilitation providers. Mentor support for OTs appeared to increase fidelity. Service managers indicated ESSVR would enhance post-stroke services.

## Conclusions

ESSVR was valued by participants and was delivered with fidelity; implementation appeared to be facilitated by mentor support for OTs.

## Introduction

Stroke is a common neurological condition resulting from a sudden interruption in blood supply to the brain caused by occlusion of a blood vessel (ischaemic stroke) or by bleeding from a blood vessel in the brain (haemorrhagic stroke) [1]. Stroke is associated with significant long-term disability. In the United Kingdom (UK) over 100,000 new strokes occur annually [2]. UK Stroke Registry data suggests a quarter of all strokes occur in adults of working age but fewer than half can expect to have returned to work at one year [3]. The COVID19 pandemic exacerbated this problem, with one study indicating people with a disability were 1.5 times more likely to be unemployed [4]. Stroke affects people at an individual and social level with reduced quality of life and poorer psychological and social outcomes. Economically there is an 18% reduction in income for individuals and lost productivity for the economy exceeding £1.6 billion per year in the UK [5]. Work is therefore an important health outcome for individuals, health services and the state [6, 7].

In the UK the National Clinical Guideline for Stroke [8] and the National Stroke Service Model [9] identify the need for Vocational Rehabilitation (VR), which is defined as a process comprising job preparation, work return, job retention or, where appropriate, planned withdrawal from work. Data from the Sentinel Stroke National Audit Programme (SSNAP) for England and Wales, indicated only 7.4% of all post-acute services audited in 2021 were commissioning a VR service [10]. In addition, just half of the 20 Integrated Stroke Delivery Networks, covering all National Health Service (NHS) regions in England and Wales, were providing VR to all stroke survivors who needed it. Three quarters of these services reported no set schedule for VR delivery prompting SSNAP report authors to call for investigation into whether VR was being delivered in appropriate doses [10].

Several studies have identified four categories of predictors of successful return to work (RTW); biological, social, psychological and environmental [11]; therefore, a multifaceted approach is required. Longitudinal data from studies of severely disabled stroke survivors from 7 countries found that access to VR varied widely between countries (24%-100%), as did RTW rates (11%-43%) [12, 13]. However, these findings were usually based on RTW data gathered at 6- or 12-months post-stroke, whereas stroke survivors often take longer to RTW [12, 13]. In addition, employers in some countries (e.g. Sweden and Australia) are

compensated when supporting stroke survivors to RTW, with VR often funded by government schemes [14, 15]. In contrast, UK stroke survivors can only access VR through the NHS unless they, or their employer, have private medical insurance or employment-related occupational health services. In addition to variability in existing VR provision, there is currently limited evidence from stroke specific systematic reviews and intervention studies to guide delivery of effective VR services to support RTW post-stroke. There is some evidence to suggest early, multidisciplinary, stroke specific interventions which cross service boundaries and include work-ability assessment, employer engagement and workplace accommodations are associated with higher RTW rates [13, 16]. Although limited confidence can be placed in these findings, they echo studies of other disabling conditions suggesting that early VR provision may support and sustain RTW post-stroke [11]. There is also consistent qualitative evidence of unmet need, poor co-ordination of services and time limited support post-stroke [5, 17, 18]. The RETurn to work After stroKE (RETAKE) trial was developed to address gaps in the existing evidence related to the provision of effective VR early after stroke [19].

RETAKE was a multi-centre individual patient randomised controlled trial (RCT) (Trial registration number: ISRCTN 12464275). It aimed to determine the clinical and cost-effectiveness of Early Stroke Specialist Vocational Rehabilitation (ESSVR) in addition to usual NHS rehabilitation on stroke survivors' RTW at 12 months post-randomisation, compared to NHS rehabilitation alone [19]. ESSVR combines occupational therapy (OT) with case-coordination and is delivered in the community over a period of up to 12 months, by a stroke- specialist OT with additional VR training. ESSVR includes the following: (a) assessing stroke impact on the person and their job; (b) educating individuals, employers, and families about the impact of stroke on work, and strategies to lessen impact (e.g., memory aids, fatigue management); (c) work preparation, and (d) liaison with employers to plan and monitor a phased and sustained RTW, where possible [19] (S1 Fig). The RETAKE trial results will be published elsewhere.

Real world implementation is central to RCT design. Therefore, there is a need to understand intervention delivery contexts and factors likely to influence intervention uptake and use [20, 21]. This is particularly important in trials of complex rehabilitation interventions, comprising multiple interacting components, and targeting different organisational levels, making them particularly challenging to implement. ESSVR crosses organisational boundaries, involves interactions between multiple stakeholders, is individually tailored, and requires behaviour change by the patient, their family and employer. Measuring intervention fidelity and gaining understanding of interactions between the intended audience and the intervention are key elements of process evaluation [21]. Fidelity measurement focuses on whether the intervention was delivered as intended (e.g., dose, content and process). Additionally, qualitative exploration of participants' experiences of an intervention and the social and structural contexts in which it was delivered is important; and enables in-depth evaluation of the adaptation or tailoring of an intervention. Understanding and reporting how the intervention is delivered (including training and support, communication, and management structures) is important for replication in clinical practice [20, 21]. This paper summarises the findings from the process evaluation embedded in the RETAKE trial.

## Methods

The process evaluation protocol was published previously [22]. The aims were to measure fidelity to the ESSVR intervention, understand the social and structural context in which the intervention was delivered and, identify factors which may influence the quality of implementation. Objectives are identified in Table 1.

**Table 1. Objectives.**

| Objectives: |
| --- |
| **1) To measure fidelity to the intervention:** |
| a. ascertain intervention dose by calculating number and length of sessions delivered. |
| b. describe level of fidelity to ESSVR intervention by coding content of Case Report Forms (CRFs) completed by RETAKE OTs. |
| c. describe content and dose of usual care (UC) and ESSVR by coding content of treatment records completed by RETAKE OTs. |
| d. observe fidelity in practice. |
| e. determine OTs competency to deliver the ESSVR intervention. |
| **2) To understand the social and structural context in which the intervention is delivered and to identify factors which may influence the quality of implementation:** |
| a. describe participating centres in terms of number and grade of qualified staff, number of support staff and caseload. |
| b. understand professionals'; experiences of being trained to deliver the intervention. |
| c. understand professionals'; experiences of delivering the intervention. |
| d. understand social and structural factors which support or act as barriers to implementation. |
| e. understand participants'; experience of being supported to RTW |
| f. Identify potential contaminants which would compromise the objective comparison between participants in intervention and UC arms. |
| g. Understand the impact of the Covid 19 pandemic and furlough scheme on stroke survivors'; work ability and the trial primary outcome (added January 2022) |

## Design

Embedded, theory-driven mixed-methods process evaluation including longitudinal case-studies, non-participant observations, semi-structured interviews and documentary analysis. The process evaluation was informed by the ESSVR intervention logic model [22] and under-pinned by Normalisation Process Theory (NPT) [23] (S1 Table) and the Conceptual Framework for Implementation Fidelity (CFIF) (S2 Fig) [24]. Development and feasibility testing of ESSVR was published previously [25]. For the ESSVR Logic Model and TIDieR description of ESSVR in the RETAKE trial see Radford et al 2021 [22]. Fig 1 outlines, recruitment, trial arm allocation, data collection by participant group and planned timing of data collection.

## PPI involvement

A Public and Patient Involvement group [26] was engaged throughout the RETAKE study. Involvement included: developing the grant proposal, advising on participant recruitment, commenting on patient-facing materials, testing baseline and follow-up questionnaires, drafting interview topic guides, commenting on data analysis, and commenting on dissemination outputs (journal manuscripts, conference abstracts and posters).

## Theoretical framework

Implementing complex interventions represents a programme change with implications for organizations, staff, and service users. Normalisation Process Theory (NPT) facilitates understanding of the dynamics of implementing, embedding, and integrating complex interventions [23]. NPT draws attention to: (i) the implementation process itself, and (ii) the organisational and structural setting in which new interventions are to be implemented. Four generative mechanisms, coherence, cognitive participation, collective action, and reflexive monitoring explain how new interventions are embedded and 'normalized' within services. These

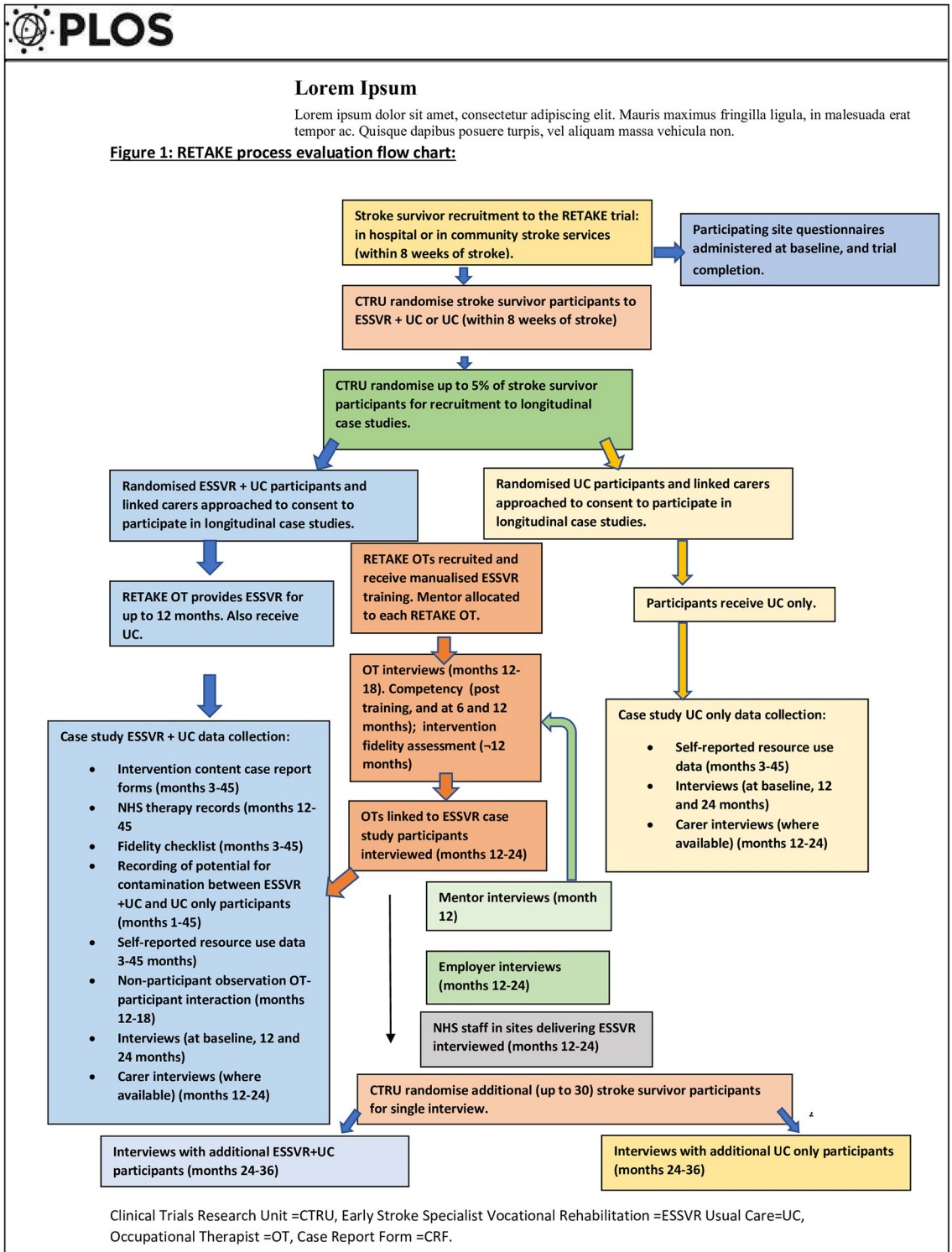

**Fig 1. RETAKE process evaluation flowchart.**

mechanisms have also been applied to analysis of patients' experiences, particularly the work that participation in treatment or rehabilitation requires as patients seek to integrate these in daily life [27, 28]. NPT informed interview schedule development and was used as a sensitising lens in qualitative data analysis.

The Conceptual Framework for Implementation Fidelity (CFIF) guided collection and analysis of quantitative data [24] CFIF outlines components and variables that make up and affect intervention fidelity and explains how they relate to each other. Adherence includes content and dose (frequency, coverage, and duration) of delivery.

## Eligibility criteria- process evaluation

RETAKE trial participants were eligible to participate in the process evaluation. Participants were recruited from four stakeholder groups: Stroke survivors, family-carers, healthcare professionals and employers (Fig 1). Recruitment commenced 1st September 2018 and ended on 31st January 2021.

Stroke survivors:

- Age ≥18 years.

- Admitted to hospital with new stroke (all severities).

- In work at stroke onset (including self-employed, paid or voluntary).

- Willing and have capacity to provide informed consent.

- Have sufficient proficiency in English to contribute to the data collection required for research.

Stroke survivors who did not intend to RTW or those with a transient ischaemic attack were excluded.

Family-carers of potential participants:

- Nominated family-carer of consenting participant.

- Willing and have capacity to provide informed consent.

- Have sufficient proficiency in English to contribute to the data collection required for research.

Also eligible were:

- OTs who delivered the ESSVR intervention and their mentors.

- NHS staff involved in management, or delivery of stroke rehabilitation in RETAKE sites.

- Employers of participants receiving ESSVR.

**Ethical approval.**   Was obtained through the East Midlands-Nottingham Research Ethics Committee (Ref: 18/EM/0019) and the National Health Service Research Authority.

## Informed consent

Following recruitment to the trial and randomisation to ESSVR plus usual care (UC) or UC only, a random sample of potential longitudinal case-study participants were identified by the CTRU (Fig 1). These participants were provided with a case-study information sheet and opportunity to ask questions. Written informed consent was obtained from all case-study

participants. Consent was reaffirmed at the start of observations and interviews. The same process was adopted for carer, employer, OT, mentor, additional stroke survivor, and NHS staff interviews. Additional consent to contact employers was requested from case-study participants before employers were contacted. Consent forms were completed and signed by participants and witnessed by Clinical Research Network staff or a Research Assistant.

## Sampling

For professional and patient interviews, purposive sampling was used to seek diversity in geographical location, staff seniority and participant sociodemographic variables.

## Data collection (Fig 1)

Data were collected at the organisational level (objectives 2a,2d,2f), intervention provider level (objectives 1a-1e and 2b-2c), and individual participant level in the ESSVR and UC arms (objective 2e). The impact of COVID19 (objective 2g) was explored at the organisational, provider and participant level.

To describe participating sites (objective 2a) researchers contacted site service managers and completed questionnaires at baseline, and at trial completion.

To ascertain intervention dose (objective 1a), OTs completed case report forms (CRFs) for each ESSVR participant for all treatment sessions (S1 File). Completed CRFs and NHS therapy records provided content descriptions of UC and ESSVR (objective 1c). To describe adherence to ESSVR (objective 1b) one ESSVR intervention record (OT therapy notes, stakeholder correspondence, and CRFs) was collected and assessed for evidence of core component delivery using an ESSVR-specific fidelity checklist (S2 Table).

OTs' competency to deliver ESSVR (objective 1e), was assessed immediately following initial training and 6-months after training using OTs' written responses to questions based on vignettes depicting novel RTW after stroke scenarios, and by assessing the clinical record of one randomly selected participant 12-months post-training.

All participant interviews were semi-structured. For case-study participants these occurred within 6 weeks of recruitment (T1), at 6 (T2) and 12 months (T3). Treating OTs, nominated carers and participants' employers were interviewed once, approximately 12 months after participant recruitment. NHS therapy records for case-study participants were reviewed (Objectives 2b-2f) approximately 12 months after recruitment. Single interviews with an additional sample of up to 30 participants (ESSVR and UC) were planned in the final year of the trial to identify any differences in delivery of ESSVR or UC on participants' experiences over time.

Interviews with OTs delivering ESSVR were planned during year one of the trial to explore views of training, the intervention, organisational and social factors influencing intervention delivery (Objectives 2b-2d and 2f). Mentors were interviewed in the same timeframe to explore their perspectives on supporting RETAKE OTs to deliver ESSVR and on organisational and social factors influencing delivery (Objectives 1e, 2c-2d and 2f). NHS staff supporting or managing OTs delivering ESSVR were also interviewed (Objectives 2a,2c-2d and 2f). Interviews were conducted by telephone by experienced qualitative researchers (KC, KP, SC, RC, CMcK) using topic guides (S2 File). Non-participant observations of OT ESSVR participant interaction in treatment sessions were planned to contribute to fidelity assessment during months 12–18 of the trial using a prompt for structured observation and unstructured fieldnotes [29].

## Quantitative data analysis: Objectives 1a- 1d

Dose, duration, and frequency of ESSVR were calculated by CTRU staff using data from CRFs and NHS therapy records. Total time spent delivering ESSVR including face-to-face contact,

liaison with the patient, employer, and other stakeholders by letter/phone, administration and travel were identified. An ESSVR-specific fidelity monitoring CRF was used to check whether the ESSVR process was followed (S2 Table) [30]. To describe UC provided during rehabilitation in community services, data were extracted from participant interviews and NHS Therapy records. Model answers developed by trainers were used to assess OTs' competence (Objective 1e) using criteria based on knowledge of the intervention process (40%), clinical reasoning (50%) and written communication (10%). Scores were mapped to a rubric identifying OTs as highly competent (≥70%), competent (50–69%) or needing additional support (≤49%) (S3 Table). A descriptive content analysis approach was utilised to determine existing stroke and VR service provision at participating sites (Objective 2a) and to identify any factors impacting on implementation of ESSVR during the trial [31]. Qualitative responses to site questionnaire items describing service provision at baseline and trial completion were reviewed by two researchers and coded to descriptive categories including: current stroke service provision (acute and community setting), evidence of VR provision, changes in stroke service or VR provision, additional service level factors which may influence implementation or delivery of ESSVR in trial sites. Researchers examined these data for any differences in site service provision between baseline and trial completion and for any factors which may have had an impact on implementation or delivery of ESSVR.

## Qualitative analysis: Objectives 1d, 2b-2f

Observational and interview data were transcribed verbatim and uploaded into QSR NVivo v12 [32]. Descriptions of UC in NHS therapy records and interview data were analysed thematically [33]. Interview data were coded in NVivo and then imported into a Framework matrix for comparison of views within and across cases and sites [31]. In parallel, case-study participant summaries were developed using the International Classification of Functioning, Disability and Health (ICF) [34] and NPT informed categories [23]. Analysis proceeded iteratively with data collection to determine whether data saturation had been achieved. Analysis followed an inductive approach of data familiarisation, line-by-line coding and development of broad themes. Themes were mapped to NPT constructs and ESSVR's core components as part of development and refinement of broader explanatory categories. To enhance reliability and encourage researcher reflexivity, each data set was independently coded by two or more RAs research assistants, these RAs met with the independent Process Evaluation lead to discuss any discrepancies in themes derived from coding. Thematic summaries, cross-referenced to the study objectives, were developed and agreed and then shared in writing and through presentation with the wider RETAKE research team, including the PPI representatives and the research active members of the Trial Management Group. These processes ensured that researchers' emerging findings and eventual conclusions were subject to robust questioning and the risk of researcher bias in data analysis and reporting minimised.

## Synthesis of qualitative and quantitative data

Quantitative and qualitative data related to intervention fidelity, case-study participant experiences of RTW support, and description of UC were synthesised at the conclusion of the trial. We compared findings from related data sets, identifying areas of dis/agreement to explain our findings.

**Impact of the COVID19 pandemic.** The pandemic impacted on UK health services and employers in public and private sectors from March 2020. Two periods of 'lockdown' occurred between March 2020 and July 2020, and October 2020 and May 2021. Lockdowns required people to work from home where possible and restricted travel. Researcher access to NHS sites

was prohibited. In late March 2020, trial recruitment was paused; it was unclear whether the trial would complete. Interviews with additional participants were brought forward (to June-September 2020) to provide as detailed a report as possible on participants' experiences of support to RTW. For these participants, corresponding interviews with OTs or employers were not planned. It was anticipated that interviewing at this time would provide some understanding of how ESSVR and UC provision and RTW, were impacted by the pandemic (Objective 2g). Final OT competency assessments were based on either the OT's last participant or the first participant randomised nine months after site opening.

## Findings

### Participating sites (Objectives 2a,2d and 2f)

Sites were purposively selected; 78 expressed interest in participating in RETAKE. Twenty were selected; four did not open to recruitment. Reasons included insufficient OTs (2 sites), problems securing research support funding (1 site) and the pandemic (all four sites). Sixteen sites from all seven English regions and Wales (S4 Table) delivered the intervention including 1 Acute, 11 Acute and Community, and 4 Community Trusts. Site-surveys at baseline indicated none had a structured early VR pathway. Five of 16 sites indicated VR provision occurred through referral to specialist centres with long waiting lists, none were stroke specific; no site provided VR of 12 months duration. End of trial surveys indicated minor changes in stroke or neurological rehabilitation services but no evidence that any site was providing an ESSVR like service. Trial contamination logs identified <10 instances of potential UC contamination risks; these were managed by the trial team and mentors. There was no evidence of RETAKE OTs treating UC participants in ways consistent with the ESSVR intervention (Objective 2f).

Sixty OTs were trained to deliver ESSVR; 48 provided most intervention sessions for between 14 and 16 participants each for 12 months. The average number of ESSVR participants treated per site was 17 (range 3–33). The average time OTs participated in RETAKE was 26 months (range 4–60). Reasons for trained OTs not delivering ESSVR included, site did not open (n = 9 OTs), and site had no participants (n = 3 OTs). Monitoring indicated 19/60 trained OTs left posts during RETAKE. Reasons included new employment (n = 9), maternity leave (n = 3), personal reasons (n = 3), other (n = 4) (31%).

### OTs experiences of training, mentoring, and competency assessment (Objectives 2b,2c, 1e)

Nineteen OTs (37%) were interviewed between May 2019 and February 2020. In the following sections we reference NPT constructs (S1 Table) in reporting how OTS experienced training and delivery of ESSVR. OTs' demographic characteristics are identified in Table 2.

**Training and mentoring: Enhancing coherence, supporting cognitive participation and collective action.** Prior to intervention delivery, OTs, the majority of whom had no prior VR experience, attended a two-day training course where they learned about ESSVR, delivering it, and completing trial documentation. Intervention manuals, including letter and report templates were provided. Following training, OTs reported varying levels of confidence in their preparedness to deliver ESSVR. To support intervention delivery, mentors, experienced in VR, were allocated to each OT. In terms of making sense of the intervention, this was facilitated by mentors and OTs meeting during training and then monthly, for hour-long group mentoring (via teleconference). Delays between training and their first participant being randomised

**Table 2. Demographic characteristics of occupational therapists interviewed.**

| Gender | |
|---|---|
| *Female* | 93% |
| | Mean (SD) |
| **Years since qualified as OT** | 18.7 (8.1) |
| **Years' experience in:** | |
| *Stroke rehabilitation* | 10.1 (8.5) |
| *Vocational Rehabilitation* | 3.8 (4.2) |
| | % |
| **Clinical Role** | |
| *NHS Occupational Therapist* | 74% |
| *NHS Occupational Therapy Team Leader* | 21% |
| *Independent practice Occupational Therapist* | 7% |
| *Senior Research Assistant* | 7% |
| **NHS Band Level**[a] | |
| *Band 6* | 63% |
| *Band 7* | 37% |
| **Education**[b] | |
| *Diploma of the College of OT* | 14% |
| *Bachelor's Degree* | 84% |
| *Relevant*[c] *Master's Degree* | 28% |

[a]: NHS staff grading for professions allied to medicine begins at Band 5 (newly qualified) higher bands indicate greater experience and more senior roles.

[b]: Data missing for 1 OT

[c]: A master's degree relating to Occupational Therapy, Health Care or Research Methods

impacted OTs' confidence in specific skills and responsibilities, as well as understanding the benefits of ESSVR,

> *We did the training in June and then[. . .]because of the issues in getting it up and running the first participant we saw was in the following April[. . .]so there was a big time lapse between the training and actually starting the intervention[. . .]That was a challenge because[. . .]it's not as fresh in your mind*
>
> *(OT for Bruce VR, site 10)*

A small number of OTs initially reported finding it difficult to participate effectively in mentoring sessions when not delivering ESSVR; some of these decided not to attend group sessions citing competing workload demands. This impacted on how OTs engaged with and thought about how they would work with ESSVR in practice. Attendance at mentoring sessions was encouraged, especially where competency assessment results suggested mentees required more support to enhance understanding.

> *I've got a new person who hasn't come into mentoring so I've emailed to say, "Can we have a ten-minute conversation?" . . . She thinks she knows what she's doing but I've just marked her paper [competency assessment] and she doesn't*
>
> *[Interview: Mentor 4].*

Overall, mentoring sessions were valued by OTs. They helped consolidate intervention knowledge and skills acquisition through informal teaching on topics e.g. employment law. Giving and receiving peer support facilitated understanding of delivering ESSVR in OT's own practice.

*I feel much more confident about how to work with employers...it's something I'll definitely continue after the study has ended [....]*

*(OT for Tom VR, site 3)*

Mentoring supported OTs who were delivering ESSVR individually at different sites, and facilitated group problem-solving, sharing best practice, and receiving feedback. This was particularly valuable where OTs were delivering ESSVR in complex cases. Mentoring provided opportunities to engage in reflexive monitoring and explore challenges and benefits in operationalising ESSVR. OTs valued being able to contact mentors between sessions, helping build confidence in intervention delivery and facilitated collective action in situations requiring more immediate attention.

*I think my issue initially was my confidence in delivering the intervention [...] It was around the employment law sorts of things [...] So, I think, as my confidence has grown, that's kind of come along.*

*(OT for Ken VR)*

For an evaluation of mentoring as part of the RETAKE trial see Craven et al, 2021 [35].

**OT competency assessment (Objective 1e).** Assessment of competence following initial training identified 15/60 (25%) OTs requiring further support who were offered additional mentoring. OT competence improved over time with far fewer requiring support (1 and 2 respectively) at assessment time-points 2 and 3 (Table 3). This was probably due to opportunities to deliver the intervention and engagement with mentoring. The development and feasibility testing of the competency assessment methods will be reported separately.

## Fidelity assessment (Objectives 1b and 1d)

Fidelity was assessed using case-review of participants' intervention records (n = 39) using a checklist to determine evidence of ESSVR delivery [30]. OTs were given an overall score (0–100%) that reflected the degree of evidence for intervention component delivery (e.g. 100% indicates an OT delivered every deliverable component). Fidelity assessment scores ranged from 30.8% to 100% with an average of 78.8% (SD: 19.2). Linear regression indicated that only the average monthly amount of mentoring the OT engaged with was associated with fidelity

**Table 3. Occupational therapist Early Stroke Specialist Vocational Rehabilitation (ESSVR) competence classification.**

| Competence Classification | Assessment Time-point | | |
|---|---|---|---|
| | **Assessment 1 (initial training)** | **Assessment 2 (refresher training)** | **Assessment 3** |
| Needs Additional Support | 25% | 2% | 3% |
| Competent | 75% | 55% | 45% |
| Highly Competent | 0% | 7% | 8% |
| Total Assessed | 60* | 38 | 34 |

*9 OTs were trained and assessed at time-point 1, their sites did not open to recruitment, they did not go on to deliver the intervention.

score (β = 0.29, 95% CI = 0.05–0.53, p < 0.05). Other attributes, including years of qualification, stroke, VR and research experience, theoretical knowledge of VR, competence following the initial training session, and total amount of mentoring received were not associated.

The pandemic restricted researcher access to all sites from late March 2020, when only 5 observations (completed over 6 months) of OTs providing ESSVR had been conducted. No further observations were possible. The five OTs observed were from different sites. All observations explored potential barriers and facilitators to ESSVR delivery and demonstrated OT adherence to ESSVR with core components relevant to the participant recorded by the researcher. Observations identified the complexity inherent in participants' adaptation to life after stroke, the wide range of post-stroke impairments experienced, challenges related to RTW and variability in family and social support. These data provided examples of how OTs tailored advice and support for ESSVR participants. Family involvement in observed sessions was evident for only 2 cases, in others it was prevented by family members' own work commitments.

## Usual care (UC) provision (Objective 1b)

Description of UC in the process evaluation relied on analysis of NHS therapy records and participants' interview transcripts. Both electronic and paper-based therapy records were difficult to access and were sometimes incomplete. During the pandemic researchers were unable to access sites. Post-pandemic there were delays in Principal Investigators, OTs and data managers providing anonymised copies of records via secure file transfer. Eleven (of 26) copies of therapy records were obtained (42%). These indicated: seven participants received community OT input (3 ESSVR; 4 UC-Only), range 2–17 sessions (mean: 6.14; SD: 5.21) and 150–895 minutes in total (data available for 4 participants; mean: 407; SD: 332.56). Four participants received community physiotherapy (2 ESSVR; 2 UC-Only) for a range of 2–13 sessions (mean: 6.25; SD: 5.32) and 150–656 minutes (data available for 3 participants; mean: 355.33; SD: 266.13). Only one participant received speech and language therapy (UC-only, two sessions). One participant (ESSVR) had early supported discharge (ESD) support for 11 sessions totalling 478 minutes, with no differentiation between professionals seen. Two participants (both ESSVR) were encouraged to access psychological services where they received support for 7 sessions (562 minutes) and 14 sessions (745 minutes) respectively. RTW was not mentioned as a rehabilitation goal in any of these records.

Self-reported UC data for case-study (n = 26) and additional interview participants (n = 18) are summarised in Table 4. Our inability to report on duration of UC in detail reflects the difficulty participants had in identifying and recalling which health professionals visited them post-stroke, how often, and for how long. However, interview data from UC and ESSVR participants consistently identified UC provision as typically of short duration (range 2–8 weeks), predominantly focused on treating physical impairments, and, for UC-only participants, was perceived as poorly coordinated with limited communication between treating therapists and between therapists and participants. UC participants commonly identified RTW as a goal, but this rarely translated into specific employment-related support. Only 1 UC participant was referred to an NHS VR service and only 5 UC participants reportedly received other work-related support. Three more UC participants saw an OT as part of ESD or community stoke rehabilitation (CSR) provision, 3 more saw their General Practitioner (GP) and 4 more saw a stroke nurse.

**Content dose, intensity and duration of ESSVR (Objective 1a).**   Trial data indicate that 97.2% (n = 307) of ESSVR participants commenced the intervention, of those 307, 41.9% (SD:23.59) were within one month of stroke These data are consistent with the reported

**Table 4. Stroke survivors' self-reported usual care provision (derived from NHS records).**

| Longitudinal Case-Study and additional interview participants (combined) | | | |
|---|---|---|---|
| **Service Accessed** | ESSVR (N = 23) | UC Only (N = 21) | Total (N = 44) |
| | % | % | % |
| **Early Supported Discharge/Community Stroke Rehabilitation (CSR)** | 78.3% | 80% | 79.5% |
| *Physiotherapy* | 52.2% | 52.4% | 52.3% |
| *Occupational Therapy* | 26.1% | 42.9% | 34.1% |
| *Speech and Language Therapy* | 26.1% | 19% | 22.7% |
| *Psychological Services* | 4.3% | 4.8% | 4.5% |
| **Health Services** | | | |
| *General Practitioner (GP)* | 17.4% | 38.1% | 27.3% |
| *Consultant* | 26.1% | 23.8% | 25% |
| *Stroke Nurse* | 4.3% | 23.8% | 13.6% |
| *Psychological Services* | 17.4% | 19% | 18.2% |
| *Physiotherapy* | 8.7% | 4.8% | 6.8% |
| *Occupational Therapy* | N/A | 9.5% | 4.5% |
| *Speech and Language Therapy* | 0% | 4.8% | 2.3% |
| **Work Related Services** | | | |
| *Employer-Related Occupational Health Service* | 17.4% | 23.8% | 920.5% |
| *Career-Specific Organisations* | 8.7% | 0% | 4.5% |
| *Job Centre* | 4.3% | 0% | 2.3% |
| *National Health Service (NHS) Vocational Rehabilitation (non-RETAKE)* | 0% | 4.8% | 2.3% |
| **Rehabilitation-Based Exercise Programmes** | 21.7% | 9.5% | 15.9% |
| **Contact with Charities** | 17.3% | 19.1% | 18.3% |

ESSVR: Early Stroke Specialist Vocational Rehabilitation

UC: Usual care provided by Early Supported Discharge (ESD) staff or community stroke rehabilitation staff

experiences of ESSVR case-study and additional interview participants and indicated early intervention in line with the core elements of ESSVR outlined in the logic model [22]. ESSVR participants were offered 10 individual OT sessions on average (SD:7.30) and trial data indicate a mean of 9.5 (SD: 7.00) sessions were attended. Mean duration of ESSVR provision was 211 days (SD:125.50).

## OTs' experiences of delivering the intervention (Objective 2c, 2d and 2g)

Where core elements of ESSVR (S1 Fig) were evident in interview data these are indicated in bold. We reference NPT (S1 Table) to illustrate barriers and facilitators to ESSVR delivery.

**Intervention delivery: Facilitators and barriers.** Intervention provision required ***individual adaptation*** to participant's needs. Collective action was facilitated when participants and other key stakeholders, including family members, employers, occupational health services, line managers, and health professionals in CSR teams, engaged in supporting OT-led RTW plans. This included allocating time and resources to RTW related activity (***communication and co-ordination***).

For participants with significant and enduring impairments including problems with walking, cognition, speech and language or vision it was more difficult and could be less feasible for OTs and employers to adapt the work situation (***mediating workplace adjustment).*** In

contrast, some OTs reported challenges working with participants who were highly self-motivated to RTW, such as self-employed participants who were often compelled to RTW for financial reasons. Their decisions did not always align with OTs' clinical judgement, and advice regarding the appropriateness of early RTW for example in relation to managing fatigue using a graded RTW process., OTs managed these cases by maintaining contact with the participant and monitoring the effectiveness of RTW over time. In other cases, participants did not give consent for OTs to contact their employer. This made it difficult for OTs to engage in the collective action needed to mediate timing of the participant's RTW, employment role or *mediate workplace adjustments*, and/or *monitor RTW.* In several cases, employers, had in-house RTW procedures. These sometimes clashed with OTs' recommendations for participants; in these cases, OTs maintained contact with and continued to support participants in their RTW.

**Differentiating ESSVR and UC service VR.** OTs reported the most notable difference was the *duration of ESSVR* which facilitated *individually tailored support*, goal setting and action planning for participants to work towards RTW. In addition, OTs demonstrated a reflexive approach, monitoring the provision of core elements of ESSVR, for example reflecting on the value of *liaising with the employer* and facilitating RTW relationships between employers and participants.

*I think without my input and without that early education and keeping things open and honest, potentially they could've been just deemed not fit to go back to work instead of talking about reasonable adjustments and what they can do and what their strengths still are*

*(OT for Tom VR site 3)*

Some OTs reported the 12-months post-randomisation discharge point was too soon for some participants who were only just returning to work. However, even in instances where a RTW was not possible, OTs felt the ESSVR process helped participants *explore alternatives* and achieve a positive outcome.

*One person who I was very involved with wasn't able to return to work, but I was quite involved in that process. I think the difference I made was, even though he couldn't go back to work, he was okay with that at the end, and it was as positive a process as it could have been*

*(OT 1, site 1).*

**Organisational and contextual factors influencing OTs delivery of the intervention.**
Many OTs reported limited prior experience with research; this contributed to difficulties when trying to make sense of delivering ESSVR. Some OTs reported frustrations, and some felt overwhelmed with completing trial documentation in addition to NHS records.

*Probably the thing that takes the most time and that's the most annoying is all the paperwork (laughs). It takes a long time, but you just fit it in amongst everything else that you have to do. You prioritise. [. . .] It's been okay. It hasn't been unmanageable.*

*(Bruce's (VR) treating OT, site 10).*

In some sites, NHS colleagues not involved in ESSVR delivery facilitated RETAKE OTs' involvement, through reviewing and then helping them manage their usual caseload. Review of workloads was evident in sites where more than one OT was involved in ESSVR provision,

providing opportunities for peer support and workload sharing. RETAKE OTs were allocated 1 new participant per month on average resulting in a typical caseload of up to 6 participants across a 12-month period; for most, this was manageable. For some OTs, treating ESSVR participants on a set day each week helped in workload management. Three OTs reported that time constraints meant they prioritised delivery of usual care OT and ESSVR and so limited their engagement with training materials and mentoring sessions; this impacted on cognitive participation and collective action.

> I guess the only thing is that you have very limited time to look at different things (training materials/intervention manual). It's the sort of thing you're having to do in your own time
>
> (OT 31, site 16).

Where OTs were employed by a different NHS Trust (healthcare provider organisation) to the Trust responsible for providing CSR, problems arose including poor, or no communication between organisations in relation to participants' care pathways, and increased travel time and costs. These issues revealed a lack of understanding and engagement of people able to facilitate service integration across different healthcare provider organisations and in terms of the need for collective action involving people working in these organisations to deliver ESSVR. In these sites, protocols or policies for cross provider working were either not set up, or not communicated effectively to treating OTs. OTs managed these challenges in different ways including reconfiguring working practices, increased reliance on telephone support and advice for participants or reducing the frequency of participant contacts.

**NHS staff experiences of managing RETAKE within sites (Objective 2d).** Semi-structured interviews with 23 NHS staff members, from 16 sites during 2020, explore the social and structural factors acting as barriers or facilitators to intervention implementation. More than one interview was conducted if initial interviews did not yield core information. In 5 sites two interviews were undertaken and in one site 3 interviews. NHS and RETAKE roles held by staff interviewed are identified in Tables 5 and 6.

Stroke survivor recruitment was affected in some sites by patients not responding to invitations to participate in the trial and in some cases low (pre-stroke) employment rates in potential patient populations. RETAKE, as do most UK trials, relied mainly on Clinical Research Network staff to recruit participants. These staff are not stroke-specialist and may have lacked the skills required to approach and recruit people with aphasia and cognitive impairments.

**Table 5. National Health Service (NHS) roles.**

| NHS roles of staff interviewed included |
| --- |
| Clinical lead / locality lead |
| Occupational Therapist (OT) (including clinical specialist) |
| Head of research |
| Research practitioner |
| Research &Development (R&D) operational lead/researcher |
| Stroke Consultant/physician |
| Clinical stroke research nurse |
| Clinical studies officer |
| Physiotherapist |
| Research delivery manager |
| Stroke coordinator and research practitioner |

**Table 6. RETAKE roles -National Health Service (NHS) staff interviewees.**

| RETAKE roles included: |
| --- |
| Principal Investigator (PI) |
| Project management/co-ordinator and set-up |
| PI and recruitment |
| Screening/recruitment |
| RETAKE Occupational Therapist (OT) providing intervention |
| PI and RETAKE OT |
| Promoting research and supporting delivery |

Varied perceptions regarding the appropriateness of recruiting patients early post-stroke may also have impacted recruitment in some sites.

> *They're still in shock that they've had a stroke, the last thing they want to think about is work. "Fine, I'm off sick. I just want to know, medically, what's going on." That's their priority*
>
> [*Site3, PI, Recruitment Staff and RETAKE OT*].

Despite these issues, most sites reportedly engaged positively with recruitment to RETAKE. Cross NHS Trust communication proved problematic in some sites adding to the workload of OTs treating participants outside their usual work areas:

> *A lot of participants randomised wouldn't have been in our group. They would have gone to other hospitals because of their GP address. So, they're additional work to what we would have normally seen, and then we've got a waiting list of our own usual care patients*
>
> (Site 7 RETAKE OT, Therapy Team Lead)

An important facilitator was the widely shared perception of the need to develop VR services in sites; RETAKE provided an opportunity to participate in research related to this. Thirteen of 16 sites reported that VR was "touched on" by acute wards or community OTs as part of general rehabilitation but lacked intensity, individually tailored content and was time limited.

> *Identifying those patients quite early on the ward, that's really important because if you look at stroke patients of working age, one of the main worries very early on is 'Will I be able to return to work?'*
>
> (Site 9, co-PI and stroke consultant).

> A big learning point for me is about that longer-term support for people and how valuable that is for stroke survivors. With VR being part of that package
>
> (Site 1, PI and RETAKE OT).

These comments highlight the need for ongoing support for stroke survivors, the early introduction of the ESSVR intervention provided stroke skilled OT support for an extended period of time, far longer than usual care services at any site.

Service managers also reported that OTs benefitted from external mentoring, upskilling in VR and having the opportunity to build their research skills.

*I'm pleased that we've been able to bring an occupational therapy trial to the Trust and to give the grassroots clinicians the opportunity to take part in a high-quality research study [. . .] to learn more about the research process and [be] part of developing the evidence base for our profession*

*(Site 16, PI).*

## Participants' experiences of support to RTW in both trial arms (Objective 2e)

Twenty-six participants were recruited to longitudinal case-studies (ESSVR n = 15 and UC n = 11) all were recruited before 31st of March 2020. Thirty-two interviews with ESSVR and 26 with UC participants were conducted between September 2018 and January 2021; some participants declined or did not respond to invitation to interviews at 6 and 12 months. Treating OTs for 11 of 15 ESSVR participants were interviewed (4 unavailable due to the pandemic). An additional 18 participants, between 5 and 24-months post randomisation, were interviewed on one occasion between July and November 2020 (ESSVR n = 8 and UC n = 10). Description of the characteristics of case-study and additional interview participants can be found in S5 Table details interviews conducted with these participants, carers and OTs. Stroke severity for case-study, participants, measured using the Oxford Cognitive Screen (OCS) and EQ5D mobility question, is reported in S6 Table. These data indicate participants experienced mild to moderate levels of post-stroke impairment with 38.5% (n = 9/26) reporting at least one impairment which may have been severe. These figures are similar to the main trial population where 34% (n = 198/583) reported at least one impairment. Mobility was the most common impairment in both the process evaluation and main trial populations at 30.8% (n = 8/26) and 16.5% (n = 96/583) respectively. Although most participants reported improvements over time, particularly in relation to physical impairments, post-stroke impairments impacted directly or indirectly on their perceived ability to RTW or to complete work as well as previously. These factors together with co- and multi-morbidities required participants to learn to cope with and adapt to changed bodies, changed functional abilities, and often changed cognitive and communication abilities. In the process evaluation several participants across groups reported that, in addition to their stroke, they experienced new, often stroke associated illness. The most reported co-morbidity was mental health problems including reduced self-confidence, low mood, work and health-related anxieties, social anxiety, and clinical depression. For some participants these conditions were disabling and impacted directly on RTW, or where they had RTW, impacted on satisfaction with the perceived quality of their work post-stroke. In most cases and to varying degrees, these factors presented as barriers to RTW. Participants faced the dual challenge of adapting to life after stroke whilst actively seeking to RTW.

In the case-study group ESSVR participants 53% (n = 8/15) had RWT at 12 months, for UC only participants 45% (n = 5/11) had RTW at 12 months. For the additional interview participants ESSVR participants 50% (n = 4/8) participants had RTW at 12 months and for UC 60% (n = 6/10) had RTW at 12 months.

In the section below we consider participants' engagement with RTW-related support. with reference to NPT's constructs. Core components of ESSVR are in highlighted in bold (S1 Fig). Anonymised data extracts are identified by pseudonym and study arm: ESSVR (VR) or usual care (UC) and site.

All participants reported receiving some community stroke rehabilitation (CSR), typically at home, and commencing 1–2 weeks after discharge from a stroke unit. Usual care CSR

duration was between 2 and 12 weeks. During this time, depending on post-stroke impairments identified, participants reported being treated by physiotherapists (PTs), occupational therapists (OTs) and, where communication difficulties were evident, by speech and language therapists (SLTs). The focus for therapy interventions during this time was typically on addressing functional limitations. Examples reported include focus on improving upper and lower limb strength and function through different forms of activity including washing and dressing, personal care, meal preparation and walking practice.

Individual stroke survivors had to make sense of their situation post-stroke and undertake RTW related activity This included adjusting to post-stroke impairments, often alongside co- or multi-morbidities. This entailed working with health professionals to understand how stroke had impacted them individually and to discuss implications for the future, including RTW plans. For ESSVR recipients this process was facilitated **early** in the post-stroke period by RETAKE OTs **providing education** about stroke and how it had affected individual participants' work abilities, to participants, their families, and their employers.*).*

For some UC participants the absence of individualised stroke-specific education provision was a concern:

*I would have liked to have known more around what the long-term effects of the stroke are. So, although there was a lot of focus on the physical aspect of it, like can you do things that you were able to before. [. . .], [but] there's some long-term things that you may be susceptible to. For example, one thing (information leaflet) said, you might get epilepsy, but if I wasn't told that I would never have known this could be a potential risk*

*(Adam (UC) Site 5).*

Accepting limitations and focussing on abilities was an important element of individual stroke survivors committing to working towards RTW). The time this process took differed for individuals. ESSVR participants and carers appreciated OTs' ***co-ordination*** and ***communication*** role:

*They (OT) coordinate the care, talk to the other therapists, make plans and have a really good relationship with [name] and keep him up to date with summaries and reports written about him [. . .] like bridge the gap between the neurologist and specialist doctors [. . .] the OT seems to be the one to get things moving*

*(Tim's Carer (VR) Site 6).*

There were some examples of therapists providing support for UC participants to understand and address problems associated with cognition, memory, decision making and visual impairment. Targeted Speech and Language provision was also evident, but this was more commonly cited as an aspect of CSR that was missing, delayed, or infrequently provided. Some participants also reported delayed or no access to psychological services:

*In hospital it was very, very good. Post-hospital, 2 weeks is not enough and what I am now tapping into with the neuropsychological support is very good, but I could have done with this 6 months ago*

*(Harry (UC) Site 14).*

In contrast, ESSVR participants benefited from sustained engagement with OTs, so that when participants were emotionally and physically ready, and where the opportunity arose,

they worked with their OT and employer to formulate RTW plans Several participants reported the OT took the lead for joint planning for RTW, work hardening/work preparation tasks, for ***individually tailored*** support including suggesting specific environmental or job-role changes in the workplace ***(accommodating stroke at work)***. In other cases, ESSVR participants reported the importance of specific on-going OT support when deciding when RTW would be appropriate, and in relation to when to seek retirement due to ill health. OTs also noted the value of providing RTW support for up to 12 months.

> *'Particularly the more complex ones, sometimes people are only just—it comes to [. . .] six months to a year and that's only when they're just kind of getting themselves in a place where they can think about work'*
>
> *(OT for Ken (VR) Site 13).*

For these ESSVR participants, OTs facilitated communication with stakeholders including employers, to gain their co-operation, with RETAKE OTs working with and **advocating** for participants.

> *I wouldn't have known where to start or how to speak. She (OT) was my hero there. [In meetings] She would just argue with them and say, "no, that's not acceptable, yes, that's acceptable." She was really fighting my case, which was great, because I wouldn't have done that myself [. . .] And they [employers] listened because they said themselves, they'd never had a stroke patient before, so they didn't know what to expect.*
>
> *(Nora (VR) Site 5).*

ESSVR participants particularly valued the RETAKE OT being available to help them understand what was happening in workplace meetings with employers:

> *I would've been stuck if [OT] hadn't been there [. . .] because there's too many questions—I wouldn't have been able to answer any of them. They [OT] understand it and they know where you're coming from. . .they listen to what you're saying*
>
> *(Dennis (VR) Site 9)*

These co-ordinated and co-operative actions were rarely reported by UC participants. Community stroke rehabilitation (CSR) appears to have helped start the process of adapting to life after stroke for UC participants. However, CSR was described as ending too early, when participants had unmet needs in coping with post-stroke impairments and their desire to RTW. A small number of UC participants reported that CSR services provided some work-preparation, signposting to local authority work-support services and leaflets from a national charity. However, many reported that RTW was either not mentioned or not followed-up:

> *Participant: I had Physio for 6 weeks afterwards; they came to my house every day. . .They used to talk to me about going back to work.*
>
> *Interviewer: Did they help you to go back to work?*
>
> *Participant: No not really*
>
> *(Pete UC Site 5).*

UC participants more commonly reported having little or no support to navigate what they perceived to be complex (state) benefits provision and more specifically in relation to RTW, to access support with contacting employers and exploring RTW opportunities. Several UC participants reported feeling abandoned once CSR was withdrawn:

*I feel like I'm left in limbo now. . .everybody's done their allotted time. . .You don't need to be left, cast aside, and wait until the next time it happens to you or you cry out for help*

(*Rory (UC) Site 2*).

*All this stuff (post-stroke impairments) is just a nightmare and there's no coordination [. . .] between the different hospital disciplines at all*

(*Harry (UC) Site 14*).

*Just because I'm on my feet, doesn't necessarily mean that I'm ready for work*

(*Larry, UC, site 5*).

*It feels like you've got only so much allotted time that they can give you*

(*Rory, UC, site 2*).

The time-limited and impairment focused CSR and the absence or loss of RTW support when CSR ended was experienced as a barrier to RTW for UC participants.

ESSVR participants worked with and were supported by treating OTs to undertake work-related tasks which were **individually tailored** to their circumstances and work-role requirements), in preparation for RTW. For ESSVR participants, OTs were instrumental in **mediating workplace adjustments** and **monitoring RTW** progress both within the workplace and through communication with participants and employer). For self-employed participants, RETAKE OTs provided *tailored* advice (e.g. on fatigue management) and *monitored RTW.* This was particularly important in cases where self-employed participants returned to work very early for financial reasons, either prior to OT involvement or against the advice of the OT.

For almost all UC arm participants these actions were undertaken independently or depended on their employer having effective Occupational Health (OH) or Human Resource services. For a small number of UC participants who worked for larger employers, including local government, the NHS, and education providers, employer-related experiences were reportedly similar to ESSVR participants. In these cases, RTW was reportedly facilitated by support from line managers or OH staff. One UC participant working in local government said:

*When I had a stroke, I knew how much full-time and part-time, sick pay, I was entitled to. I knew I would be entitled to the phased return and things like that*

(*Malcolm (UC) site 13*).

Nevertheless, interviews indicated most UC participants found RTW planning challenging in the absence of sustained OT support, advice, advocacy and co-ordinated OH and employer interaction.

ESSVR participants in working with their OT over a 12-month period, reflected on the meaning of work post-stroke both individually and with others. Whilst this motivated participants to try to RTW at their pre-stroke employment, some participants decided to work fewer

days/hours, alter roles or duties, seek a less stressful occupation, or retire ESSVR participants were supported to **explore alternatives** where RTW to the same employer and/or role was not possible. In contrast, many UC recipients reportedly felt 'abandoned' once CSR services were withdrawn early in their post-stroke journey, thus limiting their opportunity for timely or ongoing support in reviewing their RTW options.

Our data highlight some of the challenges of RTW post-stroke. ESSVR facilitated adjustment to life after stroke and provided participants with sustained stroke specialist advice and support in thinking about and planning to RTW. OTs' advocacy and proactive involvement with employers facilitated workplace adjustments and, through workplace monitoring, provided sustained support for the ESSVR participants. In contrast UC only participants typically had time limited contact with health professionals, and except in the case of those employed by large organisations with OH or HR services, had to make their own arrangements in relation to RTW planning with little or no timely or co-ordinated support from the NHS or social services.

## Discussion

This process evaluation is one of the most comprehensive theory driven mixed-methods studies evaluating an RCT of a complex intervention in stroke rehabilitation. Quantitative and qualitative data indicate it was possible for 48 OTs across 16 sites to independently and competently deliver ESSVR. There was evidence of acceptable intervention fidelity over the 12-month intervention delivery period [30]. Analysis of UC received by participants in the process evaluation revealed that CSR, whilst available to all, was of relatively short duration, was focused largely on functional improvements and was often poorly co-ordinated between health professional providers. RTW was sometimes identified as goal but often this did not lead to actions to provide ongoing support or facilitate RTW; those that were provided were often delayed for 6 months or more post-stroke. These limitations in UC provision for stroke survivors echo the findings of unmet need in several studies [5, 17, 18]. The recent Improving Primary Care After Stroke (IPCAS) research programme [18] identified several limitations in primary care provision for stroke survivors. IPCAS reported premature' withdrawal of services and a lack of follow up; that stroke survivors needs change over time, that communication could be poor between health care professionals in different sectors. Patients and carers wanted a single point of contact, and for post-stroke services to be offered pro-actively [18]. These limitations were evident in UC provision in RETAKE. Further evidence of the limitations in RTW support for stroke survivors in England was provided in the report of Patient Reported Experiences Measures for 2022/2023 in which only 28% of 3441 respondents agreed with the statement 'I felt supported with my return to work' [36, p47].

Experiences of ESSVR participants in the case-study and additional interview groups were markedly different; these participants benefitted from ongoing VR and RTW support from a stroke-specialist and ESSVR trained OT. Our findings indicate key factors underpinning the perceived value of ESSVR for stroke survivors who received this were consistent with the predicted mechanisms of action of ESSVR outlined in the pre-trial logic model [22]. Most important among these appear to be a case-coordination approach to early and individually tailored VR, written and verbal communication with participants and employers, employer engagement in planning RTW and stroke-specialist mediation in workplace adjustments. In addition, mentor support for OTs appears to have been integral to effective ESSVR delivery. A recent realist review of early intervention vocational rehabilitation (EIVR) proposed a programme theory which identified 9 mechanisms for how EIVR works, for whom and in what situations, four of which centred around employer engagement [37]. These mechanisms were all evident in the ESSVR approach and consistent with the core elements delivered by OTs in RETAKE.

We acknowledge limitations in the process evaluation, some of which were pandemic related. Only 5 non-participant observations of ESSVR delivery were possible; undertaking a wider range of these observations in additional settings at different timepoints during RETAKE would have enabled a more comprehensive review of VR and RTW support and ESSVR fidelity in practice. Access to sites to obtain and analyse more NHS therapy records would also have increased our understanding of intervention delivery. Females were underrepresented in our sample, their experiences of VR and support to RTW may have differed from those reported here. Similarly, it was not possible to include participants from all employment sectors in our sample. We acknowledge that direct reporting of employer perspectives is an important limitation in our data, but one which other researchers have also found challenging to access [38]. Another potential limitation was the risk of researcher bias towards the intervention given their familiarity with the intervention and the conduct of more than one interview with some participants. We sought to minimise this risk through the use of a structured process of data analysis and interpretation which included independent researcher and PPI member oversight and regular team review of emerging findings and their interpretation. Although this does not entirely eliminate the potential for researcher bias, we have sought to make clear the rigour of the processes we used to demonstrate the trustworthiness of our findings. Lastly, we acknowledge that we have limited evidence of the impacts of the pandemic on VR and RTW support.

The RETAKE trial results will be published in a separate publication. The overall effectiveness of ESSVR has yet to be determined. However, the process evaluation highlighted the impact of several known biopsychosocial barriers and facilitators to RTW on participant RTW outcomes at 12 months [11, 12, 15]. Barriers to RTW included mobility, cognitive and communication impairments, fatigue, visual impairment, and psychological problems including anxiety and depression. Whilst some of these factors will improve over time, particularly where specialist rehabilitation support is provided, many cannot be resolved and contribute to the high level of long-term disability seen post-stroke across the world [1–3]. Another known barrier is lack of employer or line-manager support and enterprise size [11, 15]. In the trial only 40% of participants permitted employer engagement or had an employer to engage with, this key mechanism in supporting RTW was impacted by preventing employer engagement with OTs. There are increasing numbers of younger working age stroke survivors worldwide, at the same time there is UK and international evidence of significant gaps in post-stroke rehabilitation and RTW services, particularly for this group [18, 39, 40]. Therefore some form of early, stroke-specialist VR support which includes employer involvement in RTW planning is likely to be a core requirement of integrated post-stroke services for working age stroke survivors [9, 37, 40]. The process evaluation findings showed that stroke service providers wanted to and felt it was important to provide VR for this younger cohort of stroke survivors who were already working. A recently published VR Toolkit provides stroke specific guidance and resources to facilitate VR provision in accordance with policy recommendations [41]. This further raises the profile of VR post-stroke and may encourage UK commissioners to review funding for this important element of post-stroke care.

## Conclusion

ESSVR was highly valued by participants, family carers, and OTs. In terms of integrating an ESSVR like approach in the NHS, managers' interviews revealed broad support for ESSVR which was seen as a necessary improvement on existing VR provision and in-line with the National Guideline for Stroke [8] and NHS policy directives for VR and RTW services in Integrated Stroke Delivery Networks [9] for England and Wales.

## Supporting information

**S1 Fig. Core components of ESSVR.**
(DOCX)

**S2 Fig. Assessment of fidelity and factors moderating ESSVR delivery-Conceptual Framework for Implementation Fidelity.**
(DOCX)

**S1 File. Case report form (Intervention session content).**
(PDF)

**S2 File. Interview topic guides.**
(DOCX)

**S1 Table. Normalisation Process Theory (NPT) constructs and components.**
(DOCX)

**S2 Table. Intervention fidelity checklist (Occupational therapists).**
(DOCX)

**S3 Table. Competency assessment rubric: Occupational therapists.**
(DOCX)

**S4 Table. RETAKE trial participating sites.**
(DOCX)

**S5 Table. Demographic data case-study and additional interview participants (stroke survivors).**
(DOCX)

**S6 Table. Stroke severity measured using OCS and EQ5D mobility question.**
(DOCX)

## Acknowledgments

We wish to thank the stroke survivors, family carers, employer, occupational therapists, mentors and NHS staff members who so generously gave of their time to participate in and share their experiences of this study. In addition, we would like to thank the members of the Patient and Public Involvement Group for sharing their valuable advice and expertise throughout the development, delivery and reporting of this study. We would also like to thank the members of the Trial Management and Programme Management Groups for their advice and support during the study. We would also like to thank the anonymous reviewers for their helpful comments on the manuscript.

## Author Contributions

**Conceptualization:** David James Clarke, Julie Phillips, Jain Holmes, Christopher McKevitt, Audrey Bowen, Kate Radford.

**Data curation:** David James Clarke, Katie Powers, Diane Trusson, Kristelle Craven, Alexandra Wright-Hughes, Rachel Chambers.

**Formal analysis:** David James Clarke, Katie Powers, Diane Trusson, Kristelle Craven, Christopher McKevitt, Alexandra Wright-Hughes, Rachel Chambers.

**Funding acquisition:** Christopher McKevitt, Audrey Bowen, Caroline Leigh Watkins, Amanda J. Farrin, Tracey Sach, Kate Radford.

**Investigation:** David James Clarke, Katie Powers, Diane Trusson, Kristelle Craven, Jain Holmes, Christopher McKevitt, Caroline Leigh Watkins, Amanda J. Farrin, Alexandra Wright-Hughes, Rachel Chambers, Kate Radford.

**Methodology:** David James Clarke, Katie Powers, Diane Trusson, Kristelle Craven, Julie Phillips, Jain Holmes, Christopher McKevitt, Audrey Bowen, Amanda J. Farrin, Rachel Chambers, Kate Radford.

**Project administration:** David James Clarke, Katie Powers, Kate Radford.

**Supervision:** David James Clarke, Christopher McKevitt, Kate Radford.

**Validation:** David James Clarke, Katie Powers, Diane Trusson, Kristelle Craven, Julie Phillips, Jain Holmes, Christopher McKevitt, Audrey Bowen, Caroline Leigh Watkins, Amanda J. Farrin, Alexandra Wright-Hughes, Tracey Sach, Rachel Chambers.

**Writing – original draft:** David James Clarke.

**Writing – review & editing:** David James Clarke, Katie Powers, Diane Trusson, Kristelle Craven, Julie Phillips, Jain Holmes, Christopher McKevitt, Audrey Bowen, Caroline Leigh Watkins, Amanda J. Farrin, Alexandra Wright-Hughes, Tracey Sach, Rachel Chambers, Kate Radford.

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
