## [Decision Letter · Decision Letter 0]

11 Jun 2024

PONE-D-24-17536The RETurn to work After stroKE (RETAKE) trial: findings from a mixed-methods process evaluation of the Early Stroke Specialist Vocational Rehabilitation (ESSVR) intervention.PLOS ONE

Dear Dr. Clarke,

Thank you for submitting your manuscript to PLOS ONE. After careful consideration, we feel that it has merit but does not fully meet PLOS ONE’s publication criteria as it currently stands. Therefore, we invite you to submit a revised version of the manuscript that addresses the points raised during the review process.

 In particular, the detail in the methods might be reduced and instead more detail included in the results section. This would support the discussion/conclusions on the benefits of the program. Please submit your revised manuscript by Jul 26 2024 11:59PM. If you will need more time than this to complete your revisions, please reply to this message or contact the journal office at plosone@plos.org. Please include the following items when submitting your revised manuscript:A rebuttal letter that responds to each point raised by the academic editor and reviewer(s). You should upload this letter as a separate file labeled 'Response to Reviewers'.A marked-up copy of your manuscript that highlights changes made to the original version. You should upload this as a separate file labeled 'Revised Manuscript with Track Changes'.An unmarked version of your revised paper without tracked changes. You should upload this as a separate file labeled 'Manuscript'.If applicable, we recommend that you deposit your laboratory protocols in protocols.io to enhance the reproducibility of your results. Protocols.io assigns your protocol its own identifier (DOI) so that it can be cited independently in the future. For instructions see: https://journals.plos.org/plosone/s/submission-guidelines#loc-laboratory-protocols. Additionally, PLOS ONE offers an option for publishing peer-reviewed Lab Protocol articles, which describe protocols hosted on protocols.io. Read more information on sharing protocols at https://plos.org/protocols?utm_medium=editorial-email&utm_source=authorletters&utm_campaign=protocols.

We look forward to receiving your revised manuscript.

Kind regards,

Kathleen Bennett

Academic Editor

PLOS ONE

Journal Requirements:

"This study was funded by the NIHR HTA programme (ref: 15/130/11). The views expressed herein are those of the authors, not necessarily the NIHR, the Department of Health and Social Care or the NHS."

3. Please expand the acronym “NIHR HTA” (as indicated in your financial disclosure) so that it states the name of your funders in full.

5. Please upload a copy of Figure 1, to which you refer in your text on page 8. If the figure is no longer to be included as part of the submission please remove all reference to it within the text.

Reviewers' comments:

Reviewer's Responses to Questions

**Comments to the Author**

1. Is the manuscript technically sound, and do the data support the conclusions?

Reviewer #1: Yes

Reviewer #2: Partly

2. Has the statistical analysis been performed appropriately and rigorously? 

Reviewer #1: Yes

Reviewer #2: Yes

3. Have the authors made all data underlying the findings in their manuscript fully available?

Reviewer #1: Yes

Reviewer #2: No

4. Is the manuscript presented in an intelligible fashion and written in standard English?

Reviewer #1: Yes

Reviewer #2: Yes

5. Review Comments to the Author

Reviewer #1: Very well written paper.

Process evaluation of the ESSVR intervention as part of the RETAKE trial.

Abstract is clear and concise.

Introduction is well organised and rationale for study is presented.Methods of process evaluation are very detailed.

Limitations are acknowledged.

All supplemental files are well presented and easy to follow

Some minor amendments

Could the authors include a brief anymore details description of the intervention if possible.

Supplemental file 3 refers to policy, what policy is this?

Supplemental file 1 logic model, could family support be a factor to consider for the person returning to work after stroke?

Overall very well written paper, very detailed information presented and easy to follow.

Reviewer #2: This paper outlines an extensive process evaluation, components of which are the subject of other publications. There is no doubt that this study has rigorous methodological design, though there is some indication that it wasn’t possible to implement all elements of evaluation as intended (COVID-19 being one reason).

There is a detailed methods section (page 6-14 of manuscript). The authors methodically outline each component of the process evaluation, with some results, and then refer to earlier publication for full results. The main suggestion is to reduce the detail in the methods and expand on the detail in the results, so that the results better support the discussion/conclusion about the benefit of the program. This needs to be read as a stand-alone paper.

Specific feedback as follows:

Title: Clear from the title it is a process (not outcome) evaluation. Sets the scene for study type well.

Abstract: Well constructed. Comprehensive. Logical. Easy to follow

Introduction:

• Client group is established but is there a need to define stroke?

• Ties literature into the problem statement well.

• Line 112 – “ .. but limited stroke specific qualitative evidence informs the delivery of effective VR services.” The lead up to this sentence all makes sense and is building an effective argument / statement about the lack of descriptive literature for RTW interventions, however it is hard to make sense of this particular sentence.

Methods:

Figure 1 Process Evaluation flowchart did not seem available, which is the central diagram

PPI involvement:

Line 162: “A Public and Patient Involvement group was involved throughout the RETAKE study. Involvement included:…”. To reduce the repeated use of “involve” in this sentence, suggest “A Public and Patient Involvement group was engaged throughout the RETAKE study, to: …”.

Data collection (figure 1):

Line 227 : “We collected data at the …”. To avoid the use of “we”, suggest “Data were collected at the… ”

Impact of COVID19 pandemic

Line 310 and 341: – again, consider re-wording to avoid the use of “we”.

Content dose, intensity and duration of ESSVR (Objective 1a):

Line 454: “Trial data indicate that 97.2% (n=307) of ESSVR participants commenced the intervention, with a mean of 41.9% (SD:23.59) within the first month after their stroke”. This reads that 97.2% commenced ESSVR within 1 month of stroke, so then what does the 41.9% represent?

Is this correct ? : “Of the 307 participants that commenced the intervention, 41.9% (SD:23.59) were within one month of stroke”. (suggesting re-wording to be clearer)

Intervention delivery: facilitators and barriers:

Line 467: “Interactional and skillset workability in collective action…”. This is not familiar terminology. Will the broader readership know what this means? It seems to have been introduced for the first time at this stage and is not relatable. I can see the authors have referenced NPT constructs, but the reader is required to refer to this firstly, in order to understand the commentary. It is suggested the authors manage this translation from theory and insert the practise-oriented results. This issue aside, the authors have captured some important barriers – continued impairments, blocks to liaising with employers, workplace in-house arrangements etc.

Line 511 – “Interactional workability and contextual integration…” – refer to comment above

Line 523 – the description of OTs working across different trusts is not relatable to an international audience, and whilst meaningful to the local study participants, may not have value in this publication.

Line 549 – “These staff are not stroke-specialist and may not have lacked the skills…”. Should this be “may have lacked…”

Line 586 - 31/03/2020 – should this date be written out in full?

Line 599: “In the case-study group ESSVR plus UC participants 53% (n= 8/15) had RWT at 12 months,

for UC only participants 45% (n=5/11) had RTW at 12 months. For the additional interview participants ESSVR plus UC participants 50% (n=4/8) participants had RTW at 12 months and for UC 60% (n= 6/10) had RTW at 12 months.”

This is difficult to wade through. It is established earlier that ESSVR participants also receive UC, therefore, can they be known as “ESSVR participants” in this section?

Again in the analysis of participant’s experiences the authors refer to NPT theory, eg:

Line 608: “Individual stroke survivors had to make sense of their situation post-stroke and undertake RTW related activity (coherence: individual specification)”. There doesn’t seem value in including the NPT theory correlate in brackets.

Apart from this, the points made (particularly in bolding the ESVR elements) are all important.

Discussion:

Summarises main findings well and ties back to the literature well.

Line 754 – 759 : The discussion presents the RTW outcomes for the study, which is new information introduced at the discussion. This is reported in another publication and does not seem linked to the current report. Suggest removing this.

Conclusion – states that the intervention was highly valued by all parties. There doesn’t seem to be any evidence of the employer experience, so they should be removed from this list of satisfied parties.

The conclusion to the paper has good relevance to clinical practise (health professionals in stroke rehabilitation).

6. PLOS authors have the option to publish the peer review history of their article (what does this mean?). If published, this will include your full peer review and any attached files.

Reviewer #1: No

Reviewer #2: No

---

## [Author Response · Author response to Decision Letter 0]

12 Aug 2024

PONE-D-24-17536

The RETurn to work After stroKE (RETAKE) trial: findings from a mixed-methods process evaluation of the Early Stroke Specialist Vocational Rehabilitation (ESSVR) intervention.

Revision requested Responses

Editorial revisions 

We have reviewed the author guidance, amended the manuscript accordingly, and revised file names in the resubmission accordingly.

Thank you for stating the following financial disclosure: 

"This study was funded by the NIHR HTA programme (ref: 15/130/11). The views expressed herein are those of the authors, not necessarily the NIHR, the Department of Health and Social Care or the NHS."

We have added the statement: "The funders had no role in study design, data collection and analysis, decision to publish, or preparation of the manuscript." to our cover letter as suggested.

Please expand the acronym “NIHR HTA” (as indicated in your financial disclosure) so that it states the name of your funders in full.

We have added the full definition of NIHR HTA to our cover letter as suggested.

Please note that in order to use the direct billing option the corresponding author must be affiliated with the chosen institute. Please either amend your manuscript to change the affiliation or corresponding author, or email us at plosone@plos.org with a request to remove this option. 

We have changed the corresponding author to: Professor Kathryn Radford.

Professor Radford is employed by the University of Nottingham.

Please upload a copy of Figure 1, to which you refer in your text on page 8. If the figure is no longer to be included as part of the submission, please remove all reference to it within the text. 

We apologise for this oversight. Figure 1 is an important element of the manuscript and is now included in the submission.

Please include captions for your Supporting Information files at the end of your manuscript, and update any in-text citations to match accordingly. Please see our Supporting Information guidelines for more information: 

We have provided captions in a list as indicated, these captions now match the in text citations for these files.

We have reviewed the reference list and removed references to papers which are either in development or are awaiting review. These has resulted in the removal of 6 references. We have listed the references we have removed in the cover letter as requested.

Reviewer 1 comments 

Some minor amendments

Thank you for your feedback on the manuscript.

-Could the authors include a brief anymore details description of the intervention if possible.

We have considered this suggestion and added a small amount of additional text to the introduction. Being mindful of the length of the paper we refer to core components of ESSVR which are listed in the logic model (Supplementary File 1) and the protocol for the RETAKE process evaluation (Ref 22) which provides a detailed description of the intervention using the TIDieR checklist.

Supplemental file 1 logic model, could family support be a factor to consider for the person returning to work after stroke?

Family support is a potentially supporting factor in RTW. The ESSVR intervention seeks to actively engage family members in supporting stroke survivors and recognises the likely support needs of family members too. In the process evaluation we sought to understand the involvement of family members. Work commitments for some family members of case study participants impacted on involvement, but our data suggests family member support was an important factor for many stroke survivors. The logic model we present in the manuscript is the pre-trial version which informed the process evaluation. This is likely to be revised post-publication of the trial results. The revised version will take account of influencing factors identified in the trial and process evaluation findings. The inclusion of family response in the individual outcomes element of the logic model is being considered.

Supplemental file 3 refers to policy, what policy is this?

This is one of the moderators identified in the CFIF model and, as intended by the authors Carroll et al (2007), relates to the clarity and specificity guidelines or policy which potentially influences implementation fidelity. In the case of RETAKE we focused on the National Clinical Guideline for Stroke (2019, updated 2023) and the NHS England Stroke Service Model (2021) both of which make specific reference for the kind of RTW support which should be offered to stroke survivors in the post-acute care period. These policy guidelines could have influenced the kind of usual care services that stroke survivors received.

Reviewer 2 

 Thank you for your feedback on the manuscript, we found the comments and suggestion most helpful in completing the revisions.

There is a detailed methods section (page 6-14 of manuscript). The authors methodically outline each component of the process evaluation, with some results, and then refer to earlier publication for full results. The main suggestion is to reduce the detail in the methods and expand on the detail in the results, so that the results better support the discussion/conclusion about the benefit of the program. This needs to be read as a stand-alone paper.

We accept the comment that there could be reduction in the text detailing the methods and further detail in the results sections. We have revised the methods and results sections to take account of this comment. In these revisions we sought to retain sufficient detail to allow replication of the process evaluation methods employed.

We were somewhat unclear about the reviewer’s comment that we refer to earlier publication for full results. We wish to be clear that there is no single earlier publication reporting the results of the process evaluation. However, we acknowledge that we refer to previously published papers relating to the development of a fidelity checklist (Ref 30) and more detailed reporting on embedding mentorship to support intervention implementation (ref 35), within the process evaluation. We had made reference to additional papers which have been submitted for publication but remain in review at this time. These papers will, i) report the main trial results, ii) explore how the OT competency assessment methodology was developed and evaluated, iii) focus on exploration and comparison of individual case examples from the longitudinal case study group participants, iv) report the health economic evaluation, which was a separate study. As indicated above and in the response to the editor’s comments, we have removed reference to these papers from the manuscripts but identified that they will be published elsewhere. We acknowledge that their prior inclusion in this manuscript was unhelpful and may have implied prior reporting of the findings of the process evaluation. Providing more detailed results for each of these elements (excluding the health economic study) would significantly increase the word count and make the current manuscript unwieldy. Nonetheless, we accept that adding some more detail on our findings, in the core areas of the process evaluation would enhance this manuscript. We have added to the manuscript as indicated in the track changes and clean copies of the revised manuscript.

We agree the process evaluation findings manuscript should stand alone and, with the revisions made in line with reviewers’ comments, we now believe it does so. 

Introduction:

• Client group is established but is there a need to define stroke?

• Line 112 – “ .. but limited stroke specific qualitative evidence informs the delivery of effective VR services.” The lead up to this sentence all makes sense and is building an effective argument / statement about the lack of descriptive literature for RTW interventions, however it is hard to make sense of this particular sentence.

We have added a brief definition of stroke to the beginning of the introduction 

We accept this comment. We have inserted new text as follows:

Line 116: There is consistent qualitative evidence of unmet need, poor co-ordination of services and time limited support post-stroke [5,17-18]. The RETurn to work After stroKE (RETAKE) trial was developed to address these gaps in the existing evidence related to the provision of effective VR early after stroke. [19]

Methods:

Figure 1 Process Evaluation flowchart did not seem available, which is the central diagram

We apologise for this oversight. Figure 1 is an important element of the manuscript and is now included in the submission.

PPI involvement:

Line 162: “A Public and Patient Involvement group was involved throughout the RETAKE study. Involvement included:…”. To reduce the repeated use of “involve” in this sentence, suggest “A Public and Patient Involvement group was engaged throughout the RETAKE study, to: …”.

We have amended this sentence as suggested.

Data collection (figure 1):

Line 227 : “We collected data at the …”. To avoid the use of “we”, suggest “Data were collected at the… ”

We have amended the text as suggested.

Impact of COVID19 pandemic

Line 310 and 341: – again, consider re-wording to avoid the use of “we”.

We have amended the text as suggested.

Content dose, intensity and duration of ESSVR (Objective 1a):

Line 454: “Trial data indicate that 97.2% (n=307) of ESSVR participants commenced the intervention, with a mean of 41.9% (SD:23.59) within the first month after their stroke”. This reads that 97.2% commenced ESSVR within 1 month of stroke, so then what does the 41.9% represent?

Is this correct ? : “Of the 307 participants that commenced the intervention, 41.9% (SD:23.59) were within one month of stroke”. (suggesting re-wording to be clearer)

We agree that this sentence lacked clarity and have reworded the text as suggested.

Intervention delivery: facilitators and barriers:

Line 467: “Interactional and skillset workability in collective action…”. This is not familiar terminology. Will the broader readership know what this means? It seems to have been introduced for the first time at this stage and is not relatable. I can see the authors have referenced NPT constructs, but the reader is required to refer to this firstly, in order to understand the commentary. It is suggested the authors manage this translation from theory and insert the practise-oriented results. This issue aside, the authors have captured some important barriers – continued impairments, blocks to liaising with employers, workplace in-house arrangements etc.

Thank you for this comment. The issue of how NPT is referenced in the text is a challenge in many papers. We have reviewed the two sections in which we refer to NPT (OT Intervention delivery-Barriers and Facilitators and Participants’ experiences). In these sections we have removed reference to individual components associated with NPT’s core constructs (e.g. interactional and skill set workability). In these instances, we have added or removed text, where appropriate, to clarify our explanation of the work of intervention delivery or the experiences of ESSVR participants.

Line 511 – “Interactional workability and contextual integration…” – refer to comment above

Text amended, see above.

Line 523 – the description of OTs working across different trusts is not relatable to an international audience, and whilst meaningful to the local study participants, may not have value in this publication.

We understand the point being made here. We have added to this section by clarifying that NHS Trusts are healthcare provider organisations and used this form of words as part of the explanation of the problems OTs encountered when working across different healthcare provider organisations.

Line 549 – “These staff are not stroke-specialist and may not have lacked the skills…”. Should this be “may have lacked…”

Yes, this was a typographical error.

Line 586 - 31/03/2020 – should this date be written out in full?

Now written in full.

Line 599: “In the case-study group ESSVR plus UC participants 53% (n= 8/15) had RWT at 12 months,

for UC only participants 45% (n=5/11) had RTW at 12 months. For the additional interview participants ESSVR plus UC participants 50% (n=4/8) participants had RTW at 12 months and for UC 60% (n= 6/10) had RTW at 12 months.”

This is difficult to wade through. It is established earlier that ESSVR participants also receive UC, therefore, can they be known as “ESSVR participants” in this section?

We have revised this section to remove the reference to ‘plus UC’.

Again in the analysis of participant’s experiences the authors refer to NPT theory, eg:

Line 608: “Individual stroke survivors had to make sense of their situation post-stroke and undertake RTW related activity (coherence: individual specification)”. There doesn’t seem value in including the NPT theory correlate in brackets.

Thank you for this comment. We understand the point being made. We have amended the participant experiences section in line with the OT section, retaining broad reference to NPTs constructs, so as to be consistent with our intention to present a theory driven process evaluation. However, in line with your comment we have removed the reference to individual components of NPTs constructs such as individual specification. 

Discussion:

Line 754 – 759 : The discussion presents the RTW outcomes for the study, which is new information introduced at the discussion. This is reported in another publication and does not seem linked to the current report. Suggest removing this.

We accept that this statement introduces new information in the discussion section of the manuscript. Its inclusion here was to enter into some discussion about the possible reasons for the finding of no significant difference between the RTW rates in the trial arms. However, we recognise there is limited opportunity in this manuscript to review these issues. We have now removed the section reporting the headline trial result.

Conclusion – states that the intervention was highly valued by all parties. There doesn’t seem to be any evidence of the employer experience, so they should be removed from this list of satisfied parties.

 We have removed the reference to employers from the first line of the conclusion.

---

## [Editor Report · Decision Letter 1]

23 Aug 2024

PONE-D-24-17536R1The RETurn to work After stroKE (RETAKE) trial: findings from a mixed-methods process evaluation of the Early Stroke Specialist Vocational Rehabilitation (ESSVR) intervention.PLOS ONE

Dear Dr. Clarke,

Thank you for submitting your manuscript to PLOS ONE. After careful consideration, we feel that it has merit but does not fully meet PLOS ONE’s publication criteria as it currently stands. Therefore, we invite you to submit a revised version of the manuscript that addresses the points raised during the review process. There remain a few comments to address. In particular the use of reporting checklists for qualitative /mixed methods studies which were not applied.

We look forward to receiving your revised manuscript.

Kind regards,

Kathleen Bennett

Academic Editor

PLOS ONE

Journal Requirements:

Additional Editor Comments:

1. PLOS ONE considers qualitative and mixed-methods studies for publication. We recommend that authors use the COREQ checklist, or other relevant checklists listed by the Equator Network, such as the SRQR, to ensure complete reporting (http://journals.plos.org/plosone/s/submission-guidelines#loc-qualitative-research). In general, we would expect qualitative studies to include the following: 1) defined objectives or research questions; 2) description of the sampling strategy, including rationale for the recruitment method, participant inclusion/exclusion criteria and the number of participants recruited; 3) detailed reporting of the data collection procedures; 4) data analysis procedures described in sufficient detail to enable replication; 5) a discussion of potential sources of bias; and 6) a discussion of limitations.

2. The legends for some of the tables require further detail - the tables should be self-explanatory/standalone

3. Descriptive content analysis is mentioned in the quantitative analysis methods but no details provided. Also, how is this presented in the results? Cell counts in some of the tables is very small (e.g. 1 or 2) - should avoid presenting small numbers for ethical concerns (potential identification).
---

## [Author Response · Author response to Decision Letter 1]

10 Sep 2024

PONE-D-24-17536

The RETurn to work After stroKE (RETAKE) trial: findings from a mixed-methods process evaluation of the Early Stroke Specialist Vocational Rehabilitation (ESSVR) intervention.

Revision requested Responses

Editorial revisions 

We have reviewed the author guidance, amended the manuscript accordingly, and revised file names in the resubmission accordingly.

Thank you for stating the following financial disclosure: 

"This study was funded by the NIHR HTA programme (ref: 15/130/11). The views expressed herein are those of the authors, not necessarily the NIHR, the Department of Health and Social Care or the NHS."

We have added the statement: "The funders had no role in study design, data collection and analysis, decision to publish, or preparation of the manuscript." to our cover letter as suggested.

Please expand the acronym “NIHR HTA” (as indicated in your financial disclosure) so that it states the name of your funders in full.

We have added the full definition of NIHR HTA to our cover letter as suggested.

Please note that in order to use the direct billing option the corresponding author must be affiliated with the chosen institute. Please either amend your manuscript to change the affiliation or corresponding author, or email us at plosone@plos.org with a request to remove this option. 

We have changed the corresponding author to: Professor Kathryn Radford.

Professor Radford is employed by the University of Nottingham.

Please upload a copy of Figure 1, to which you refer in your text on page 8. If the figure is no longer to be included as part of the submission, please remove all reference to it within the text. 

We apologise for this oversight. Figure 1 is an important element of the manuscript and is now included in the submission.

Please include captions for your Supporting Information files at the end of your manuscript, and update any in-text citations to match accordingly. Please see our Supporting Information guidelines for more information: 

We have provided captions in a list as indicated, these captions now match the in text citations for these files.

We have reviewed the reference list and removed references to papers which are either in development or are awaiting review. These has resulted in the removal of 6 references. We have listed the references we have removed in the cover letter as requested.

Reviewer 1 comments 

Some minor amendments

-Could the authors include a brief anymore details description of the intervention if possible.

- Supplemental file 1 logic model, could family support be a factor to consider for the person returning to work after stroke?

-Supplemental file 3 refers to policy, what policy is this?

Thank you for your feedback on the manuscript.

We have considered this suggestion and added a small amount of additional text to the introduction. Being mindful of the length of the paper we refer to core components of ESSVR which are listed in the logic model (Supplementary File 1) and the protocol for the RETAKE process evaluation (Ref 22) which provides a detailed description of the intervention using the TIDieR checklist.

Family support is a potentially supporting factor in RTW. The ESSVR intervention seeks to actively engage family members in supporting stroke survivors and recognises the likely support needs of family members too. In the process evaluation we sought to understand the involvement of family members. Work commitments for some family members of case study participants impacted on involvement, but our data suggests family member support was an important factor for many stroke survivors. The logic model we present in the manuscript is the pre-trial version which informed the process evaluation. This is likely to be revised post-publication of the trial results. The revised version will take account of influencing factors identified in the trial and process evaluation findings. The inclusion of family response in the individual outcomes element of the logic model is being considered.

This is one of the moderators identified in the CFIF model and, as intended by the authors Carroll et al (2007), relates to the clarity and specificity guidelines or policy which potentially influences implementation fidelity. In the case of RETAKE we focused on the National Clinical Guideline for Stroke (2019, updated 2023) and the NHS England Stroke Service Model (2021) both of which make specific reference for the kind of RTW support which should be offered to stroke survivors in the post-acute care period. These policy guidelines could have influenced the kind of usual care services that stroke survivors received.

Reviewer 2 

 Thank you for your feedback on the manuscript, we found the comments and suggestion most helpful in completing the revisions.

There is a detailed methods section (page 6-14 of manuscript). The authors methodically outline each component of the process evaluation, with some results, and then refer to earlier publication for full results. The main suggestion is to reduce the detail in the methods and expand on the detail in the results, so that the results better support the discussion/conclusion about the benefit of the program. This needs to be read as a stand-alone paper.

We accept the comment that there could be reduction in the text detailing the methods and further detail in the results sections. We have revised the methods and results sections to take account of this comment. In these revisions we sought to retain sufficient detail to allow replication of the process evaluation methods employed.

We were somewhat unclear about the reviewer’s comment that we refer to earlier publication for full results. We wish to be clear that there is no single earlier publication reporting the results of the process evaluation. However, we acknowledge that we refer to previously published papers relating to the development of a fidelity checklist (Ref 30) and more detailed reporting on embedding mentorship to support intervention implementation (ref 35), within the process evaluation. We had made reference to additional papers which have been submitted for publication but remain in review at this time. These papers will, i) report the main trial results, ii) explore how the OT competency assessment methodology was developed and evaluated, iii) focus on exploration and comparison of individual case examples from the longitudinal case study group participants, iv) report the health economic evaluation, which was a separate study. As indicated above and in the response to the editor’s comments, we have removed reference to these papers from the manuscripts but identified that they will be published elsewhere. We acknowledge that their prior inclusion in this manuscript was unhelpful and may have implied prior reporting of the findings of the process evaluation. Providing more detailed results for each of these elements (excluding the health economic study) would significantly increase the word count and make the current manuscript unwieldy. Nonetheless, we accept that adding some more detail on our findings, in the core areas of the process evaluation would enhance this manuscript. We have added to the manuscript as indicated in the track changes and clean copies of the revised manuscript.

We agree the process evaluation findings manuscript should stand alone and, with the revisions made in line with reviewers’ comments, we now believe it does so. 

Introduction:

• Client group is established but is there a need to define stroke?

• Line 112 – “ .. but limited stroke specific qualitative evidence informs the delivery of effective VR services.” The lead up to this sentence all makes sense and is building an effective argument / statement about the lack of descriptive literature for RTW interventions, however it is hard to make sense of this particular sentence.

We have added a brief definition of stroke to the beginning of the introduction 

We accept this comment. We have inserted new text as follows:

Line 116: There is consistent qualitative evidence of unmet need, poor co-ordination of services and time limited support post-stroke [5,17-18]. The RETurn to work After stroKE (RETAKE) trial was developed to address these gaps in the existing evidence related to the provision of effective VR early after stroke. [19]

Methods:

Figure 1 Process Evaluation flowchart did not seem available, which is the central diagram

We apologise for this oversight. Figure 1 is an important element of the manuscript and is now included in the submission.

PPI involvement:

Line 162: “A Public and Patient Involvement group was involved throughout the RETAKE study. Involvement included:…”. To reduce the repeated use of “involve” in this sentence, suggest “A Public and Patient Involvement group was engaged throughout the RETAKE study, to: …”.

We have amended this sentence as suggested.

Data collection (figure 1):

Line 227 : “We collected data at the …”. To avoid the use of “we”, suggest “Data were collected at the… ”

We have amended the text as suggested.

Impact of COVID19 pandemic

Line 310 and 341: – again, consider re-wording to avoid the use of “we”.

We have amended the text as suggested.

Content dose, intensity and duration of ESSVR (Objective 1a):

Line 454: “Trial data indicate that 97.2% (n=307) of ESSVR participants commenced the intervention, with a mean of 41.9% (SD:23.59) within the first month after their stroke”. This reads that 97.2% commenced ESSVR within 1 month of stroke, so then what does the 41.9% represent?

Is this correct ? : “Of the 307 participants that commenced the intervention, 41.9% (SD:23.59) were within one month of stroke”. (suggesting re-wording to be clearer)

We agree that this sentence lacked clarity and have reworded the text as suggested.

Intervention delivery: facilitators and barriers:

Line 467: “Interactional and skillset workability in collective action…”. This is not familiar terminology. Will the broader readership know what this means? It seems to have been introduced for the first time at this stage and is not relatable. I can see the authors have referenced NPT constructs, but the reader is required to refer to this firstly, in order to understand the commentary. It is suggested the authors manage this translation from theory and insert the practise-oriented results. This issue aside, the authors have captured some important barriers – continued impairments, blocks to liaising with employers, workplace in-house arrangements etc.

Thank you for this comment. The issue of how NPT is referenced in the text is a challenge in many papers. We have reviewed the two sections in which we refer to NPT (OT Intervention delivery-Barriers and Facilitators and Participants’ experiences). In these sections we have removed reference to individual components associated with NPT’s core constructs (e.g. interactional and skill set workability). In these instances, we have added or removed text, where appropriate, to clarify our explanation of the work of intervention delivery or the experiences of ESSVR participants.

Line 511 – “Interactional workability and contextual integration…” – refer to comment above

Text amended, see above.

Line 523 – the description of OTs working across different trusts is not relatable to an international audience, and whilst meaningful to the local study participants, may not have value in this publication.

We understand the point being made here. We have added to this section by clarifying that NHS Trusts are healthcare provider organisations and used this form of words as part of the explanation of the problems OTs encountered when working across different healthcare provider organisations.

Line 549 – “These staff are not stroke-specialist and may not have lacked the skills…”. Should this be “may have lacked…”

Yes, this was a typographical error.

Line 586 - 31/03/2020 – should this date be written out in full?

Now written in full.

Line 599: “In the case-study group ESSVR plus UC participants 53% (n= 8/15) had RWT at 12 months,

for UC only participants 45% (n=5/11) had RTW at 12 months. For the additional interview participants ESSVR plus UC participants 50% (n=4/8) participants had RTW at 12 months and for UC 60% (n= 6/10) had RTW at 12 months.”

This is difficult to wade through. It is established earlier that ESSVR participants also receive UC, therefore, can they be known as “ESSVR participants” in this section?

We have revised this section to remove the reference to ‘plus UC’.

Again in the analysis of participant’s experiences the authors refer to NPT theory, eg:

Line 608: “Individual stroke survivors had to make sense of their situation post-stroke and undertake RTW related activity (coherence: individual specification)”. There doesn’t seem value in including the NPT theory correlate in brackets.

Thank you for this comment. We understand the point being made. We have amended the participant experiences section in line with the OT section, retaining broad reference to NPTs constructs, so as to be consistent with our intention to present a theory driven process evaluation. However, in line with your comment we have removed the reference to individual components of NPTs constructs such as individual specification. 

Discussion:

Line 754 – 759 : The discussion presents the RTW outcomes for the study, which is new information introduced at the discussion. This is reported in another publication and does not seem linked to the current report. Suggest removing this.

We accept that this statement introduces new information in the discussion section of the manuscript. Its inclusion here was to enter into some discussion about the possible reasons for the finding of no significant difference between the RTW rates in the trial arms. However, we recognise there is limited opportunity in this manuscript to review these issues. We have now removed the section reporting the headline trial result.

Conclusion – states that the intervention was highly valued by all parties. There doesn’t seem to be any evidence of the employer experience, so they should be removed from this list of satisfied parties.

 We have removed the reference to employers from the first line of the conclusion.

---

## [Editor Report · Decision Letter 2]

13 Sep 2024

The RETurn to work After stroKE (RETAKE) trial: findings from a mixed-methods process evaluation of the Early Stroke Specialist Vocational Rehabilitation (ESSVR) intervention.

PONE-D-24-17536R2

Dear Dr. Clarke,

We’re pleased to inform you that your manuscript has been judged scientifically suitable for publication and will be formally accepted for publication once it meets all outstanding technical requirements.

Kind regards,

Kathleen Bennett

Academic Editor

PLOS ONE
---

## [Editor Report · Acceptance letter]

30 Sep 2024

PONE-D-24-17536R2 

PLOS ONE

Dear Dr. Clarke, 

I'm pleased to inform you that your manuscript has been deemed suitable for publication in PLOS ONE. Congratulations! Your manuscript is now being handed over to our production team.

Kind regards, 

on behalf of

Professor Kathleen Bennett 

Academic Editor

PLOS ONE